# Characterization of the Volatile Compounds in *Camellia oleifera* Seed Oil from Different Geographic Origins

**DOI:** 10.3390/molecules27010308

**Published:** 2022-01-04

**Authors:** Jing Wang, Xuxiao Tang, Qiulu Chu, Mengyu Zhang, Yingzhong Zhang, Baohua Xu

**Affiliations:** 1Guangdong Provincial Key Laboratory of Silviculture, Protection and Utilization, Guangdong Academy of Forestry, Guangzhou 510520, China; wangjing@sinogaf.cn (J.W.); tangxuxiao666@163.com (X.T.); rainbow84397520@163.com (M.Z.); baohuaxu@aliyun.com (B.X.); 2Jiangsu Co-Innovation Center of Efficient Processing and Utilization of Forest Resources, College of Light Industry and Food Engineering, Nanjing Forestry University, Nanjing 210037, China; chuqiulu@njfu.edu.cn

**Keywords:** *Camellia oleifera* seed oil, geographical classification, HS-SPME/GC–MS, volatile compounds

## Abstract

Volatile flavor of edible oils is an important quality index and factor affecting consumer choice. The purpose of this investigation was to characterize virgin *Camellia oleifera* seed oil (VCO) samples from different locations in southern China in terms of their volatile compounds to show the classification of VCO with respect to geography. Different samples from 20 producing VCO regions were collected in 2020 growing season, at almost the same maturity stage, and processed under the same conditions. Headspace solid-phase microextraction (HS-SPME) with a gas chromatography–mass spectrometer system (GC–MS) was used to analyze volatile compounds. A total of 348 volatiles were characterized, including aldehydes, ketones, alcohols, acids, esters, alkenes, alkanes, furans, phenols, and benzene; the relative contents ranged from 7.80–58.68%, 1.73–12.52%, 2.91–37.07%, 2.73–46.50%, 0.99–12.01%, 0.40–14.95%, 0.00–27.23%, 0.00–3.75%, 0.00–7.34%, and 0.00–1.55%, respectively. The VCO geographical origins with the largest number of volatile compounds was Xixiangtang of Guangxi (L17), and the least was Beireng of Hainan (L19). A total of 23 common and 98 unique volatile compounds were detected that reflected the basic and characteristic flavor of VCO, respectively. After PCA, heatmap and PLS-DA analysis, Longchuan of Guangdong (L8), Qingshanhu of Jiangxi (L16), and Panlong of Yunnan (L20) were in one group where the annual average temperatures are relatively low, where annual rainfalls are also low. Guangning of Guangdong (L6), Yunan of Guangdong (L7), Xingning of Guangdong (L9), Tianhe of Guangdong (L10), Xuwen of Guangdong (L11), and Xiuying of Hainan (L18) were in another group where the annual average temperatures are relatively high, and the altitudes are low. Hence, volatile compound distributions confirmed the differences among the VCO samples from these geographical areas, and the provenance difference evaluation can be carried out by flavor.

## 1. Introduction

*Camellia oleifera*, a kind of theaceous evergreen shrub tree or middle arbor, has been cultivated for more than 2300 years in central and southern regions of China that also distributes in Japan and other Southeast Asia areas [1,2]. *C. oleifera* seed oil contains squalene [3], sterol [3,4,5], tocopherol [5], polyphenols [6], and a high content (≥90%) of unsaturated fatty acids (mainly oleic acids and linoleic acid) [7,8], and plays important roles in antioxidant [9,10], anti-inflammatory [11], hepatoprotective, and gastroprotective functions [12]. Virgin *C. oleifera* seed oil (VCO) belongs to a natural product and is obtained by mechanical or physical methods without any further refining process. The VCO contains many components that are favorable in terms of health. One of the most important reasons is the volatile compounds [13] that are principally generated by fatty acid oxidation and have great concern because of their impact on the quality of foods and the sensory attributes. The specific flavor of VCO is also one of the many factors considered separately from the other edible vegetal oils.

The volatile compounds of edible oil comprise of several short-chain hydrocarbons or a minimal number of polar functional groups with good nonpolar features, such as aldehydes, alcohols, ketones, acids, furan, phenols, and esters [14]. They are commonly responsible for the characteristic flavor of edible vegetal oil, which plays a significant role in the food industry because it has prime influence on consumer choice [15,16,17]. However, in hundreds of different volatile compounds, only a small fraction actually contributes to the overall flavor [18]. Hence, the volatile profiles can be used to estimate the quality of VCO and identify the variety of *Camellia*.

In recent years, gas chromatography–mass spectrometry (GC–MS), electronic nose (E-nose), gas chromatography–olfactometry (GC–O), and so on have been widely developed in the food and pharmaceutical industry [19,20]. Among them, GC–MS analysis has the excellent ability of simultaneously separating, identifying, even quantifying the multiple volatile components [21,22].

The flavor composition and nutritional evaluation of edible plants in different growing locations has been an important research field in the past few years, such as focusing on volatile compounds, fatty acids, amino acids, polyphenols, and antioxidant activities in *Capsicum annuum* [23], *Viburnum opulus* L. [24], olive [18,25,26,27], *Taxus Baccata* L. [28], *Paeonia ostii* [29], and *Camellia sinensis* [30]. It has long been known that the volatile compounds of edible oil are related to genetic (cultivars) [31,32,33,34,35], environmental (geography [18,25,27,31], climatic conditions [27,31] and storage conditions [31]), cultivating (agronomic techniques [36] and the degree of fruit ripening [31,36]), and processing (harvesting methods [31] and processing technology [37]) factors. Therefore, geographic origin of *C. oleifera* is greatly responsible for the sensorial characteristics of VCO. Moreover, the volatile compounds of oil obtained from different production areas, under identical growth conditions, harvested at roughly equal ripeness degree, and processed in the same manner, can be characterized by different compositions and their respective concentrations. There has been increasing interest in the geographical identification of virgin plant oil, as a reliable criterion for its authentication and quality [25].

However, until now, the research on the differences of VCO flavor characteristics mainly focuses on processing treatment [1,38]. There is little information on the identification of the main odorants in China VCO from different geographic origins. Therefore, the aim of this study was to investigate the characterization of the volatile compounds extracted from VCO produced in different geographical locations of southern China by GC–MS with multivariate statistical methods, and to establish the specific volatile substances or their categories that probably affect the VCO flavor from different geographical regions.

## 2. Results and Discussion

### 2.1. Comprehensive Analysis of Volatile Compounds

Geographical factors have a strong effect on the formation of chemically volatile substances of *Camellia* oil [34], especially for virgin oil [35]. In order to evaluate the characterization of the flavor of virgin *Camellia* oleifera seed oil (VCO) from different planting locations in southern China, HS-SPME/GC–MS was performed to analyze the volatile compounds in this study. The ion total chromatogram curves of 20 VCO samples are demonstrated in Figure 1. The number and retention times of the ion chromatographic peaks represent the differences of volatile compounds of VCO. In L1 to L20 samples, L4 showed the largest number of peaks, which indicates that it might contain more quantity of volatile compounds. Moreover, L16 displayed the longest retention time span and it had an obvious abundance at 47.50 min that was hexatriacontane by system software analysis. It could be intuitively reflected from Figure 1 that there are differences in the types and quantities of volatile components of 20 VCO from different geographic regions. Hence, the number and content of volatile components of all VCO samples will be further studied.

### 2.2. Composition Analysis of Volatile Components

#### 2.2.1. Analysis of the Contents and Quantity of Volatile Compounds

Not only can threshold values of specific volatile compounds affect the flavor of edible oil to a certain extent [17], the type and content of flavor substances also play an important function in the odorant. The analysis result by HS-SPME/GC–MS shows that a total of 348 volatile compounds were identified in 20 VCO samples. According to different properties, the volatiles of VCO samples are divided into 10 chemical categories (Figure 2), namely ketones, alcohols, alkanes, esters, aldehydes, alkenes, acids, phenols, furans, and benzenes, which play significant roles in the discrimination of the VCO.

Among them, ketones (69) were the most abundant volatile compounds in this study (Figure 2A); these are formed by auto-oxidation of fatty acids, β-oxidation, and decarboxylation [39], and usually give the sweet and fruity flavor. Alcohols (57) and aldehydes (48), also as main compounds in all VCO samples, being formed by oxidative degradation of fatty acids or Strecker degradation of amino acids [38], contributed the overall odor of *Camellia* oil with large quantity; these findings are consistent with a previous report [34]. A total of 51 esters were identified that can be produced by esterification of alcohols with free small molecular fatty acids, or by enzymatic degradation of amino acids during the growth of *Camellia oleifera*. Most esters are described as having fruit and flower aromas. In addition to the above four kinds of substances, alkanes (51) and alkenes (48) were also main types of volatile components and play a key role in the overall flavor of VCO. The remaining number of acids, phenols, furans, and benzenes were 24, 9, 8, and 4, respectively. These multivariate compounds and their interactions constitute the complicated flavor of VCO.

VCO from different planting areas usually presents a unique flavor, which is closely related to the relative proportion and number of flavor substances. Hence, the relative percentage contents of VCO volatile components quantitatively calculated by peak area normalization method (Figure 2B) and the quantities (Figure 2C) are discussed in this study.

As shown in Figure 2B, the highest proportions of aldehydes and furan were detected in L19 (58.68% and 3.75%, respectively), ketones and alkenes in L8 (12.52% and 14.95%, respectively), alcohols in L11 (37.07%), acids in L4 (46.50%), esters in L3 (12.01%), and alkanes, phenols, and benzenes in L20 (27.23%, 7.34%, and 1.55%, respectively). Some predominant aldehydes. such as furfural and (Z)-13-octadecenal, had particularly high proportion, which reached 31.38% in L18 and 24.29% in L6, respectively (data not shown in figure). Octanoic acid also accounted for high levels in the L4 sample with a relative content of 22.60%, which usually has a strong fat flavor at high concentrations [34]. It has been reported that geographical origin is an important factor affecting the phenolic formation [26]. However, in this study, phenolic compounds could be found only in L8, L16, L19, and L20, and not in the other 16 samples.

From Figure 2C, the number of volatile compounds in L17 oil (88) was 1.07–1.87 times higher compared to other locations samples. It is worth mentioning that the aldehyde with the highest content was in the L19 oil, but the quantity (15) in this oil was not a maximum. The largest number of aldehydes was 24, both in L6 and L15. The highest content of ketones was in L8; nevertheless, the largest number was in L2. L11 had the highest content of alcohols, the number of which in L11 sample was six. Although the number of alcohols in L20 was also six, its content was the lowest in all samples, at only 2.91%. There is no correlation between the contents and quantity of various compounds in each sample, but they both show that the main types of volatile substances in VCO are aldehydes, ketones, alcohol, acids, esters, alkenes, and alkanes. In the study of other food flavors with regard to geographical origins, aldehydes and alcohols are considered the largest volatile profiles in olive oil with the 2-hexenal and 1-hexanol more than 50% [18,27] and 12% [25], respectively. Alkanes are the main volatile compounds in bell pepper spices [23], and acids and ketones are the main volatile compounds in *Viburnum opulus* L. fruits [24]. Compared to other oils or plants, VCO presents its own special flavor.

The formation of different odorants does not necessarily result from a high number of volatile compounds [40]. Nevertheless, our study revealed that the relative content and quantity of volatile compounds of VCO in different locations have their own characteristics. This specificity together could constitute the uniqueness of the flavor of each VCO sample. Therefore, this study will further classify VCO by multivariate statistical analysis to find the similarity and regularity of VCO from 20 geographic regions.

#### 2.2.2. Analysis of the Common and Unique Volatile Components

After analysis of all the VCO samples from different geographic regions, a total of 23 common volatile components were detected, including 12 of aldehydes, 1 of ketones, 3 of alcohols, 4 of acids, 1 of esters, 1 of alkenes, and 1 of phenols (Table 1). These compounds construct the basic flavor of VCO. Among them, decanal with sweet flavor, 2,4-decadienal with deep-fried flavor, (E)-2-decenal with fatty flavor, and 2-undecenal with fresh aldehyde flavor [41], were found in all of VCO samples and their retention indexes were approximate (1204–1311), suggesting that these flavors are highly common in VCO. It is particularly noteworthy that they are all aldehydes and the specific reason for this occurrence pattern remains to be determined. Hexanal was also a common compound of aldehydes and found in 19 VCO samples, which is consistent with the previous research [38,42]. It is produced by linoleic acid oxidation and shows high content in tea seed oil, grape seed oil, soybean oil, and corn oil [34]. The analysis of common volatile compounds confirms again that aldehydes, which universally exist in olive oil [18], are also major and important volatile compounds in VCO.

Except for aldehydes, acids were also main common volatile substances of VCO, including hexanoic acid, 4-hydroxy-butanoic acid, nonanoic acid, and octanoic acid, which is consistent with previous reports [33]. In four of the common acids, hexanoic and octanoic acid show a VCO sensory description of sweety and fatty that matches the flavor of decanal and (E)-2-decenal. Nonanoic acid is characterized by cheese and sweet that also could be identified as the volatile compounds of most fruits [38]. It is noteworthy that L8 sample had not common acids that probably implies the specific of L8.

In addition, the benzyl alcohol, 2-furan methanol and 1-heptanol as common alcohols are contributed to the overall flavor of VCO with aromatic, bitterness, and woody. The other common volatile compounds of ketones, esters, alkenes, and phenols were 5-butyldihydro-2(3H)-furanone, 3-phenyl-2-propenoic acid methyl ester, 8-methyl-1-undecene, and maltol, respectively. Maltol (3-hydroxy-2methyl-4-pyrone), which is usually found in roasted cocoa powder as the volatile flavor compound of caramel [43], was also not recognized in the L8 sample.

If the common volatile compounds show the basic flavor of VCO, the unique volatile components in each VCO sample may be one of the factors influencing the formation of characteristic flavor. Table 2 lists a total of 98 unique volatile compounds detected in VCO from different geographic regions. The sensory descriptions of the unique volatile substances are not shown in Table 2, because most of detected unique volatile compounds have only a few studies. The number of unique aldehydes, ketones, alcohols, acids, esters, alkenes, alkanes, furans, and phenols were **5**, **23**, **11**, **9**, **13**, **16**, **13**, **4**, and **4**, respectively. And the number of unique volatile components tested in L1-L20 samples were **2**, **4**, **4**, **14**, **3**, **6**, **2**, **7**, **4**, **7**, **8**, **0**, **7**, **2**, **7**, **3**, **3**, **0**, **7**, and **8**, respectively.

L4 had the largest number of unique volatile compounds, indicating that its characteristic flavor was probably different from other samples. The unique volatile compounds of L4 were mostly concentrated in acids and esters that shows the specific flavor of flowers, iris, and fruit (octanoic acid pentyl ester). Moreover, the characteristic flavor of the L8 sample was probably weak floral, woody, and slight scented with laurel oil (dodecanoic acid). The characteristic flavor of the L20 sample might be herbs (4-ethyl-2-methoxy-phenol), spicy (2-methoxy-4-methyl-phenol), and phenol odor (2-ethyl-phenol). Meanwhile, no specific compounds were detected in L12 and L18, demonstrating that the flavor of these two samples may be more popular and have few characteristics.

Therefore, the common and unique volatile compounds reflect the basic and characteristic flavor of VCO, respectively, and reveal the commonness and specificity of VCO flavor in different provenance areas, which is conducive to the establishment of VCO brands with local characteristics.

### 2.3. Multivariate Statistical Analysis

#### 2.3.1. Principal Component Analysis

To fully understand the differences of VCO among 20 important planting areas of *Camellia oleifera*, a multivariate statistical analysis is applied to reveal the distribution and relationship of their volatile components. From the score plots of the principal component analysis (PCA) for ketones, alcohols, alkanes, esters, aldehydes, alkenes, acids, phenols, furans, and benzenes as shown in Figure 3A, all samples were scattered in four parts. The distribution of variability was mainly driven by the first two principal components that accounted for 80.6% of the total variance (factor 1, 60.5%; factor 2, 20.1%). The samples that have more specific volatile substances, such as L4, L8, and L20, were farther away from the PCA center point. Therefore, it could be inferred that the 23 common (Table 1) and 98 unique (Table 2) volatile compounds determine the particularity of VCO flavor in various planting regions.

In addition, the samples in the same group are indistinguishable. The location of variables in the loading plot explains the reasons why certain observations form clusters in the score plot [26]. By the PCA of the volatile components of VCO, the L8 and L20 oils located in the left part, while the other samples were distributed around the center. It indicates that the overall volatile profile of two oils is similar but different from the other 18 location samples. In previous research on olive oil, growing locations, such as climatic [27] and pedoclimatic conditions [25], were found to affect the odorant profiles. We believe that this conclusion is also applicable to VCO study.

#### 2.3.2. Heatmap Analysis

Heatmap is another statistical method widely used in recent years. It can aggregate many data to show the results as gradual color bands to illustrate the density and frequency of the data [1]. To investigate the variable distribution among the groups based on the PCA results, heatmap analysis of the volatile components of the 20 tested oils was employed. The results are presented as a visual heat map added to the dendrogram in Figure 3B.

According to the relative contents of volatile components, the VCO samples from 20 regions were divided into two main categories, which is consistent with the results of PCA analysis. The first included the samples of L8, L16, and L20 in which the contents of phenols, alkanes, benzene, alkenes, and ketones are relatively high. In the second main category, it could be divided into two subordinate classifications. The first included the samples of L2, L4, L5, L12, L13, and L15 that the contents of ketones, acids, and esters are comparatively high. Moreover, there were two sorts in the second subordinate classification that the first included L1, L3, L14, L17, and L19 oils and the contents of esters, furans, and aldehydes are high. For the rest of the oils, the contents of esters, alcohols, alkenes, and ketones are high.

#### 2.3.3. Partial Least Squares-Discrimination Analysis

Partial least squares-discriminant analysis (PLS-DA) is a kind of clustering or separation method with two data matrices of X (explanatory dataset) and Y (explicative dataset) [44], which has been widely applied for biomarker selection in metabolomics [45]. Based on the PCA and heatmap analysis, PLS-DA was carried out to further classify 20 VCO with flavor characteristics. As shown in Figure 4A, a significant discrimination according to the data matrix of their volatile compounds of VCO can be observed by using a PLS-DA model. The cross-validated predictive capability (Q^2^ = 0.268, *p* < 0.005) indicates the model’s good feasibility. The four groups were then clustered with A group for the sample dots of pink (L8, L16, and L20), B group for red (L2, L4, L5, L12, L13, and L15), C group for green (L1, L3, L14, L17, and L19), and another D group for the sample dots of purple (L6, L7, L9, L10, L11, and L18). The A group was in the lower part, whereas the B, C, and D groups accumulated in the upper part of the picture, four of which had partial overlapping. Thus, the groupings in the scatter plot are with respect to geography.

To better understand the metabolites that affect the contribution in classification of the four groups in the PLS-DA model, values of variable importance in projection (VIP) are commonly used to calculate and identify for volatile compounds, especially in study of geographical discrimination [23]. Usually, the average VIP on a particular model is 1. When VIP exceeds 1, the variable is considered to have important function on the PLS-DA discriminant process [44,46]. In Figure 4B, the variables of the PLS-DA model of volatile data with VIP values greater than 1.0 included acids, alcohols, aldehydes, and alkanes in descending order. It indicates that these compounds are potential markers for the clustering and classification of the four groups in PLS-DA score plot.

## 3. Material and Methods

### 3.1. Materials

*Camellia oleifera* fruits or seeds were collected from 20 geographical location in southern China, including Guangdong, Hunan, Jiangxi, Hainan Province, and Guangxi Zhuang Autonomous Region. In each area, 3 or 4 *C. oleifera* samples were collected. All *C. oleifera* fruits were harvested at almost the same maturity stage during the crop season 2020. The sampling sites were primarily selected based on the geographical location, cultivars, altitude, annual average temperature, annual rainfall, annual sunshine, and major climate types of the region, which may influence the fruit ripening capacity and quality potential (Table 3).

### 3.2. Oil Extraction

After sun exposure and manual shelling, only the *C. oleifera* seeds with no infection or those that were physically damaged were obtained from fruits and put into the oven at 75 °C for hot air drying until constant weight [47,48]. Then, the seeds were crushed and transferred to a BOZY-01G screw press from Hanhuang Electric Appliance Technology Co., Ltd. (Zhejiang, China). After the screw-pressing process, the crude oil was centrifuged at 10,000 rpm for 10 min at 4 °C. Finally, the supernatant oil as virgin *C. oleifera* seed oil (VCO) was kept in brown glass bottles in a cool place until analysis.

### 3.3. Volatile Compounds Analysis

#### 3.3.1. HS-SPME

The volatile compounds analysis of VCO was based on the previous flavor research [18] at which point we further optimized them [49]. The 20 mL samples of VCO were put into 40 mL headspace vials, which were hermetically sealed with silicone pad. The headspace vials with oil were allowed to equilibrate for 10 min at room temperature. The target volatile organic substances of the samples were extracted for 33 min at 40 °C using a headspace solid phase microextraction manual sampler injection handle (Shanghai Anpu Experimental Technology Co., Ltd., China) with 50/30 μm Divinylbenzene/Carboxen/Polydimethylsiloxane (DVB/CAR/PDMS) solid-phase microextraction (SPME) fiber. The volatile compounds of oils were desorbed by directly inserting SPME fiber for 3 min into the injection port of gas chromatography maintained at 250 °C.

#### 3.3.2. GC–MS Analysis

The Shimadzu QP2020 Gas Chromatography–Mass Spectrometer system (GC–MS) equipped with a flame ionization detector (FID) was used to analyze volatile components. The MS signal for the identification was simultaneously obtained by the GC system and the odor characteristics of each compound were detected by a sniffing port. An SH-Rxi-5Sil MS capillary column (30 m × 0.25 mm × 0.25 μm) was used for analysis in GC system. The carrier gas was helium with a purity of 99.999%.

Operation conditions were as follows: 40 °C of the oven temperature of SH-Rxi-5Sil MS capillary column, 250 °C of the temperature of injection block, splitless of injection mode, 50 of split ratio, 1 min of injection time, linear speed of flow control mode, 36.1 cm/s of line speed, 49.7 kpa of pressure, 54.1 mL/min of total flow rate, 1 mL/min of column flow rate, and 3 mL/min of purge flow rate.

The temperature first increased to 40 °C, remaining for 1.5 min, and then to 230 °C at 4 °C/min with a final hold at 230 °C for 3 min. The GC–MS interface and ionization source temperatures were set at 280 and 230 °C, respectively. The solvent delay time was 1 min. The total program analysis time was 52 min.

#### 3.3.3. Qualitative Analysis

The qualitative analysis of the volatile compounds was processed by GC–MS software analysis to identify unknowns with the ability of peak picking, peak deconvolution, and mass spectra comparison. Automatic integration was used by peak area with 200 of peak number and 2 s of half peak width. The volatile compounds were identified by comparing the mass spectra with the mass spectrometry libraries of the National Institute of Standards and Technology (NIST17). The similarities of the components were all above 80%. The relative percentage content of each volatile compound in VCO was calculated by area normalization method according to the peak area.

### 3.4. Statistical Analysis

All experiments were performed in triplicate. Data and charts were processed using Microsoft Office 2010. Before the multivariate chemometric methods were applied to the original data, the data were standardized by SPSS 19.0 (IBM, SPSS version 19 IBM Corp., Armonk, NY, USA). Principal component analysis (PCA) was employed to identify the main factors controlling the composition. Heatmap analysis was performed by TB Tool software (version v1.099) to analyze the relationship with the volatile compounds and the different geographical area. Partial least squares-discrimination analysis (PLS-DA) was applied to classify all of the samples according to their volatiles using MetaboAnalyst 3.5.

## 4. Conclusions

Different from other popular vegetable oils, most virgin *Camellia oleifera* seed oils (VCO) from distinct producing areas have unique flavor. Therefore, this study is a first in exploring the effect of volatile compounds on the geographical discrimination of VCO from the perspective of flavor. Ten chemical categories of volatile compounds, including aldehydes, ketones, alcohols, acids, esters, alkenes, alkanes, furans, phenols, and benzenes, were detected, both number and contents of which in each sample were distinguished. A total of 23 common and 98 unique volatile compounds were determined that cause the basic and characteristic flavor of VCO from different geographic origins, respectively. The 20 main producing regions of *C. oleifera* in southern China were classified into four groups according to the VCO flavor. The regions of Longchuan in Guangdong (L8), Qingshanhu in Jiangxi (L16), Panlong in Yunnan (L20) where plants of the cultivars of *C. oleifera* Abel. belonged to the same category, scatter plots of which were in the lower part in partial least squares-discrimination analysis compared to other groups. From the perspective of growing environments, the annual average temperatures in these three locations are relatively low (lowest at 13 °C) and the annual rainfalls are also low (1035–1650 mm). Moreover, another group contained six planting areas of Guangning in Guangdong (L6), Yunan in Guangdong (L7), Xingning in Guangdong (L9), Tianhe in Guangdong (L10), Xuwen in Guangdong (L11), and Xiuying in Hainan (L18) where the annual average temperatures are relatively high (highest at 29 °C) but the altitudes are low (≤1000 m). The influence of geography and climate on VCO flavor is highly complicated. The geographical characterization of the other two groups were not found in our study, probably due to there being some unknown factors affecting their classification. VCO from different parts of the region have their own defining characteristics that can be used in the authentication studies and geographical classification of China *Camellia* oils further to promote the development and utilization of VCO as an edible oil.

## Figures and Tables

**Figure 1 molecules-27-00308-f001:**
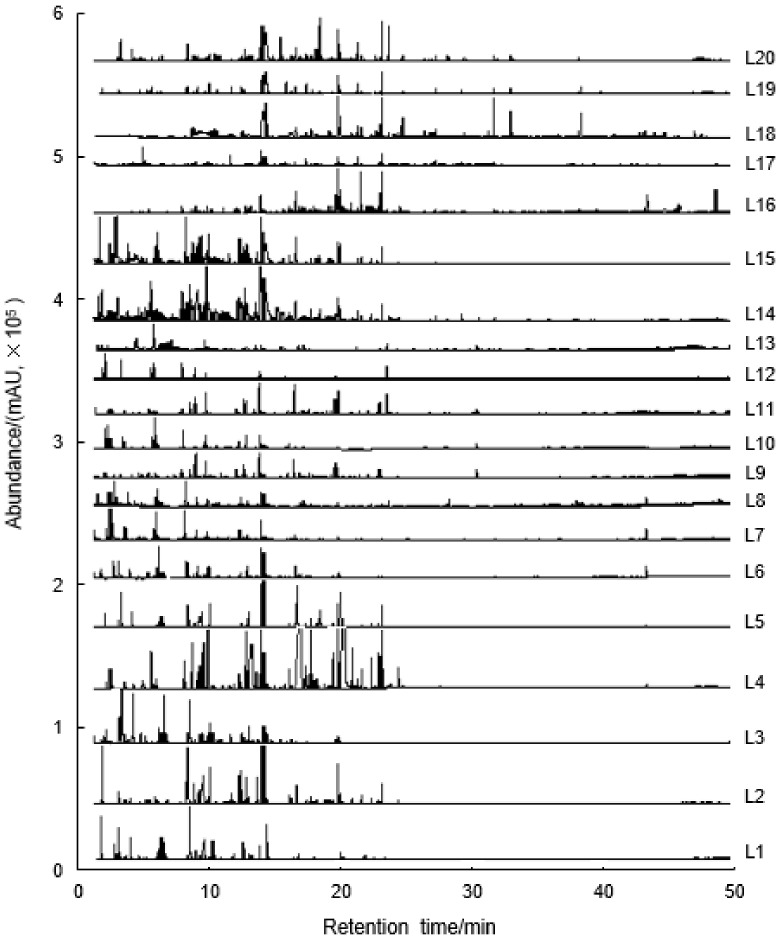
Total ion chromatograms of volatile compounds of virgin *Camellia oleifera* seed oil (VCO) from 20 geographic regions.

**Figure 2 molecules-27-00308-f002:**
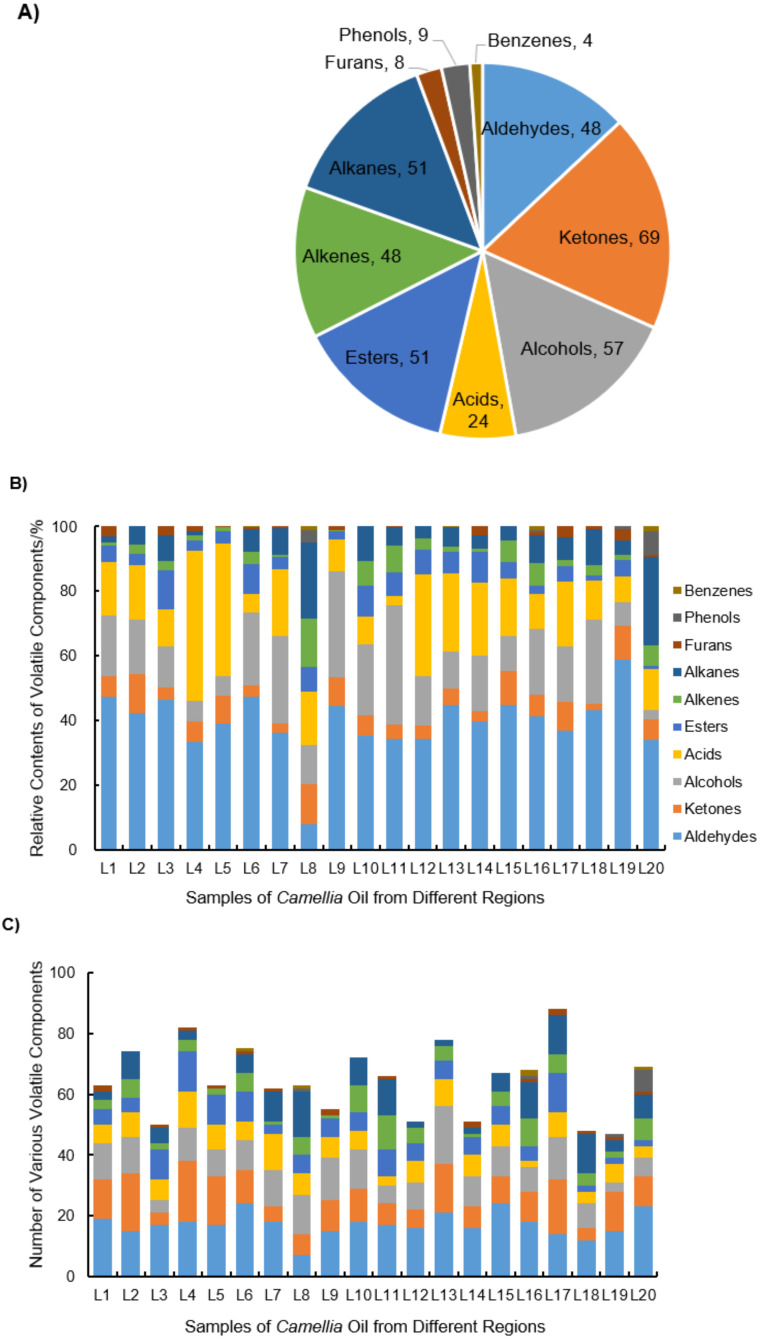
The composition and content of volatile compounds of VCO from 20 geographic regions. (**A**) The number of ketones, alcohols, alkanes, esters, aldehydes, alkenes, acids, phenols, furans, and benzenes in all *Camellia oleifera* seed oils. (**B**) The relative contents of 10 chemical categories of volatiles in VCO samples from 20 regions. (**C**) The number of various volatile components in VCO samples from 20 regions.

**Figure 3 molecules-27-00308-f003:**
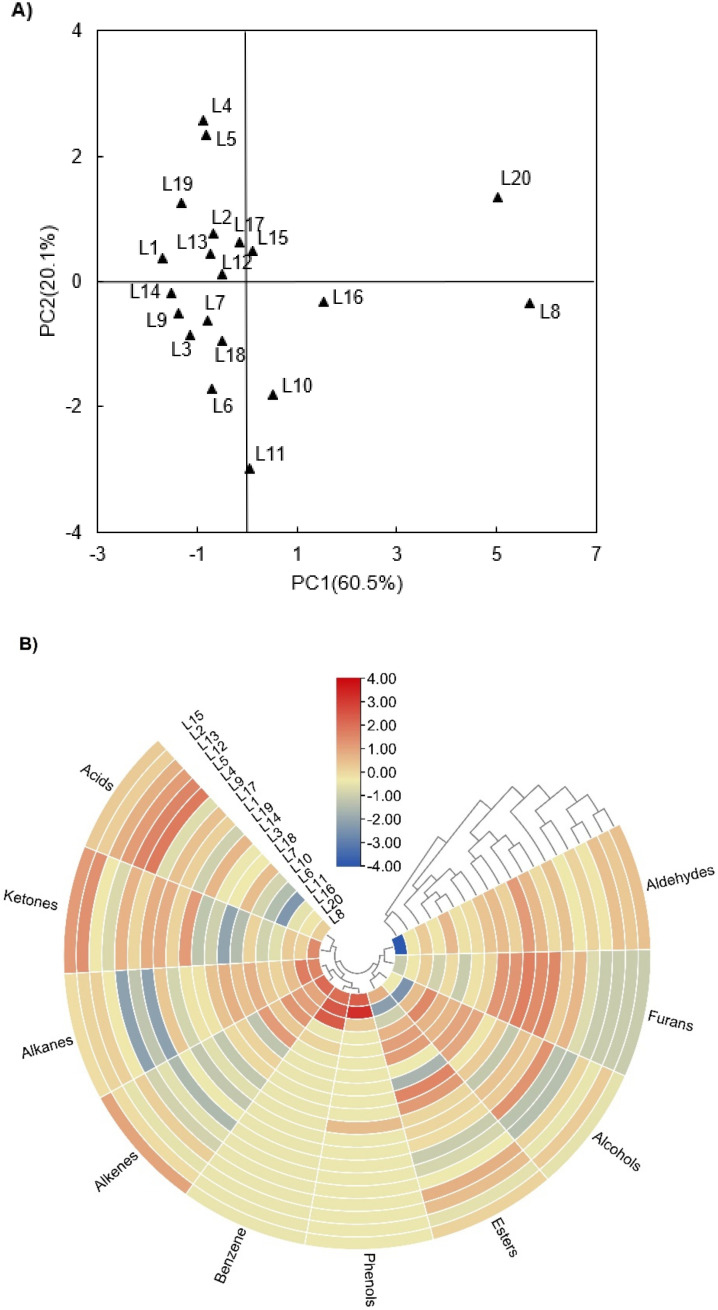
Principal component analysis and heatmap analysis of volatile compounds of VCO from 20 geographic regions. (**A**) Score plot of principal component analysis of all volatile compounds data of *Camellia oleifera* seed oil. (**B**) Heat map analysis of *Camellia* oil using the main types of f volatile composition data.

**Figure 4 molecules-27-00308-f004:**
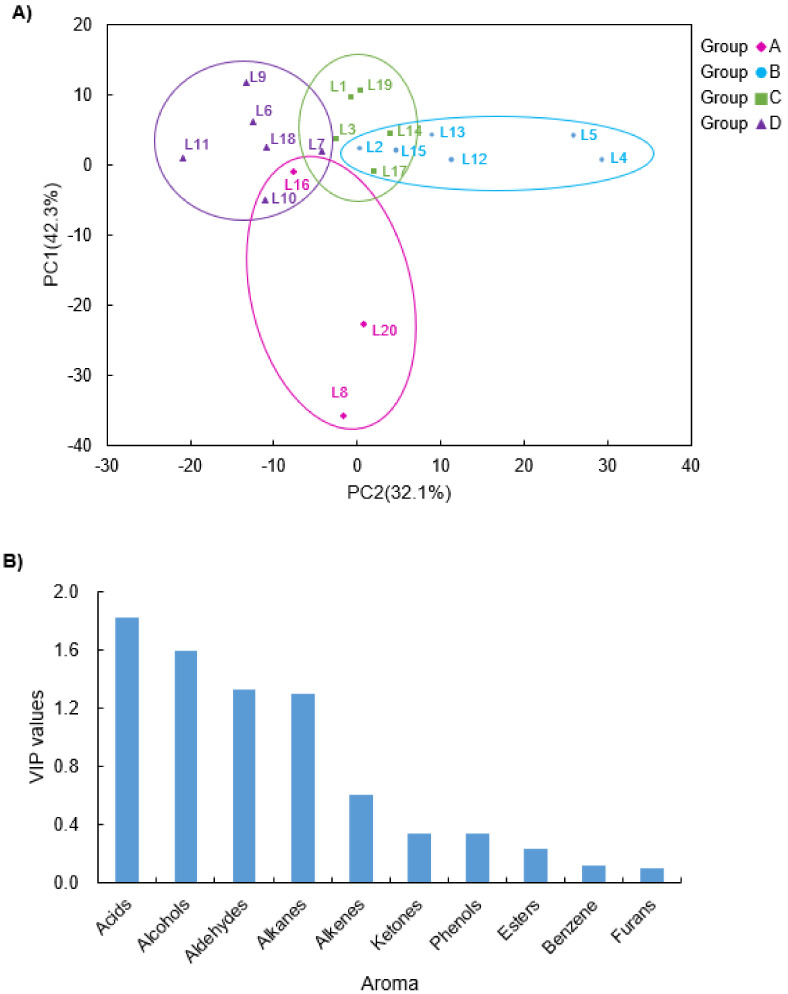
Partial least squares-discrimination analysis of volatile compounds of VCO from 20 geographic regions. (**A**) Score plot of partial least squares-discrimination analysis of all volatile compounds data of *Camellia oleifera* seed oil. (**B**) The variables important in the projection scores of 10 chemical categories of volatile compounds.

**Table 1 molecules-27-00308-t001:** Information of common volatile compounds of VCO from 20 geographic regions.

NO.	Volatile Compound	CAS	Formula	Retention Index	Sensory Descriptions ^a^	Unrecognized Samples
Aldehydes						
**1**	Decanal	112-89-0	C_10_H_20_O	1204	Sweet, waxy	——
**2**	2,4-Decadienal	2363-88-4	C_10_H_16_O	1220	Deep-fried	——
**3**	(E)-2-Decenal	3913-81-3	C_10_H_18_O	1212	Fatty, green	——
**4**	2-Undecenal	2463-77-6	C_11_H_20_O	1311	Strong fresh aldehyde	——
**5**	Hexanal	66-25-1	C_6_H_12_O	806	Cut grassy, apple	L8
**6**	(E)-2-Nonenal	18829-56-6	C_9_H_16_O	1112	Green, fatty	L8
**7**	Octanal	124-13-0	C_8_H_16_O	1005	Vanilla, orange	L19
**8**	Heptanal	111-71-7	C_7_H_14_O	905	Green plant, oily	L8, L18
**9**	Nonanal	124-19-6	C_9_H_18_O	1104	Grassy, Almond	L11, L14
**10**	(Z)-2-Heptenal	57266-86-1	C_7_H_12_O	913	Oxidised, pungent	L14, L15, L19
**11**	Furfural	98-01-1	C_5_H_4_O_2_	831	Almond	L8, L15, L16, L20
**12**	(E)-2-Octenal	2548-87-0	C_8_H_14_O	1013	Green, floral	L3, L8, L16, L19
Ketones						
**13**	γ-Octanoic lactone	104-50-7	C_8_H_14_O_2_	1184	Peach, coconut, oatmeal bread	L3, L6, L18
Alcohols						
**14**	1-Heptanol	111-70-6	C_7_H_16_O	960	Fresh, woody	L15, L16, L19
**15**	2-Furan methanol	98-00-0	C_5_H_6_O_2_	885	Bitterness	L8, L15, L16, L20
**16**	Benzyl alcohol	100-51-6	C_7_H_8_O	1036	Aromatic	L5, L8, L12, L18, L19
Acids						
**17**	Octanoic acid	124-07-2	C_8_H_16_O_2_	1173	Oily, fatty	L8
**18**	Hexanoic acid	142-62-1	C_6_H_12_O_2_	974	Sweet, pungent	L8, L20
**19**	Nonanoic acid	112-05-0	C_9_H_18_O_2_	1272	Cheese, sweet	L8, L9, L18
**20**	4-Hydroxybutanoic acid	591-81-1	C_4_H_8_O_3_	1018	Buttery, rancid	L8, L15, L16, L20
Ester						
**21**	Methyl cinnamate	103-26-4	C_10_H_10_O_2_	1267	Cherry, balsamic flavor	L4, L9, L14, L20
Alkenes						
**22**	8-Methyl-1-undecene	74630-40-3	C_12_H_24_	1140	NF	L8, L10, L18, L20
Phenols						
**23**	Maltol	118-71-8	C_6_H_6_O_3_	1063	Caramel	L2, L4, L8, L17, L20

Note: ^a^, Sensory descriptions were obtained from Fenaroli’s Handbook of Flavor Ingredients [41]. NF, not found.

**Table 2 molecules-27-00308-t002:** Information of unique volatile compounds of VCO from 20 geographic regions.

NO.	Volatile Compound	CAS	Formula	Retention Index	Similarity (%)	Recognized Sample
Aldehydes						
**1**	2,3-Dihydro-4-carboxaldehyde	37414-43-0	C_10_H_10_O	1348	81	L6
**2**	3-Hydroxy-4-methoxy-benzaldehyde	621-59-0	C_8_H_8_O_3_	1392	90	L20
**3**	(Z)-13-Octadecenal	58594-45-9	C_18_H_34_O	2007	86	L6
**4**	13-Tetradecenal	85896-31-7	C_14_H_26_O	1591	80	L10
**5**	(Z)-4-Undecenal	68820-32-6	C_11_H_20_O	1311	88	L15
Ketones						
**6**	γ-Butyrolactone	96-48-0	C_4_H_6_O_2_	825	85	L19
**7**	Cyclopentadecanone	502-72-7	C_15_H_28_O	1970	85	L6
**8**	1,4-Cyclooctanedione	55794-45-1	C_8_H_12_O_2_	1302	84	L4
**9**	2,3-Dihydro-3,5-dihydroxy-6-methyl-4(4H)-pyranone	29446-10-4	C_6_H_8_O_4_	1269	92	L19
**10**	Dihydro-5-methyl-3(2H)-furanone	34003-72-0	C_5_H_8_O_2_	821	83	L11
**11**	3-Nonanone	925-78-0	C_9_H_18_O	1053	93	L13
**12**	4-Dodecanone	6137-26-4	C_12_H_24_O	1350	83	L15
**13**	2,5-Dimethyl-4-hydroxy-3(2H)-furanone	3658-77-3	C_6_H_8_O_3_	1022	93	L19
**14**	5-Hexyldihydro-2(3H)-furanone	706-14-9	C_10_H_18_O_2_	1383	85	L10
**15**	1-Hydroxy-2-butanone	5077-67-8	C_4_H_8_O_2_	798	85	L9
**16**	9-Hydroxy-2-nonanone	25368-56-3	C_9_H_18_O_2_	1295	83	L2
**17**	1-Indanone	83-33-0	C_9_H_8_O	1218	81	L20
**18**	5-Isopropylfuran-2(3H)-one	1315481-67-4	C_7_H_10_O_2_	956	80	L2
**19**	4-Methyl-cyclopentadecanone	34894-60-5	C_16_H_30_O	2031	85	L6
**20**	4-Methyl-2-hexanone	105-42-0	C_7_H_14_O	789	90	L20
**21**	4-Methyl-2-oxepanone	2549-60-2	C_7_H_12_O_2_	1126	89	L7
**22**	4-Methyl-4-penten-2-one	3744-02-3	C_6_H_10_O	721	86	L19
**23**	(E)-3-Octen-2-one	18402-82-9	C_8_H_14_O	960	95	L17
**24**	3-Pentylcyclopentanone	85163-13-9	C_10_H_18_O	1145	88	L13
**25**	Solavetivone	54878-25-0	C_15_H_22_O	1645	85	L11
**26**	Tetrahydro-6-pentenyl-pyran-2-one	25524-95-2	C_10_H_16_O_2_	1205	82	L5
**27**	2-Tridecanone	593-08-8	C_13_H_26_O	1449	94	L15
**28**	3,3,6-Trimethyl-1,5-heptadien-4-one	546-49-6	C_10_H_16_O	1042	84	L4
Alcohols						
**29**	[S-(R*,R*)]-2,3-Butanediol	5341-95-7	C_4_H_10_O_2_	743	96	L14
**30**	Diglycerol	59113-36-9	C_6_H_14_O_5_	1504	93	L16
**31**	Glycerine	56-81-5	C_3_H_8_O_3_	967	96	L16
**32**	1,5-Heptadiene-3,4-diol	51945-98-3	C_7_H_12_O_2_	1040	91	L13
**33**	(Z)-9-Hexadecen-1-ol	10378-01-5	C_16_H_32_O	1862	93	L10
**34**	6-Methyl-5-hepten-2-ol	1569-60-4	C_8_H_16_O	924	88	L5
**35**	6-Methyl-2-hepten-4-ol	153665-39-5	C_8_H_16_O	923	88	L5
**36**	2-Methyl-2-nonen-1-ol	43161-19-9	C_10_H_20_O	1243	89	L4
**37**	2-Octanol	123-96-6	C_8_H_18_O	1060	97	L9
**38**	E-2-Tetradecen-1-ol	75039-86-0	C_14_H_28_O	1664	91	L8
**39**	2,4-Undecadien-1-ol	59376-58-8	C_11_H_20_O	1373	92	L15
Acids						
**40**	2-Decenoic acid	3913-85-7	C_10_H_18_O_2_	1380	98	L4
**41**	Dodecanoic acid	143-07-7	C_12_H_24_O_2_	1570	97	L8
**42**	Heptanoic acid	111-14-8	C_7_H_14_O_2_	1074	97	L13
**43**	2-Heptenoic acid	18999-28-5	C_7_H_12_O_2_	1081	96	L4
**44**	(E)-3-Hexenoic acid	1577-18-0	C_6_H_10_O_2_	982	95	L19
**45**	(E)-2-Methyl-2-butenoic acid	80-59-1	C_5_H_8_O_2_	860	92	L8
**46**	2-Methyl-propanoic acid	79-31-2	C_4_H_8_O_2_	711	89	L8
**47**	(E)-2-Octenoic acid	1871-67-6	C_8_H_14_O_2_	1181	93	L4
**48**	Tetradecanoic acid	544-63-8	C_14_H_28_O_2_	1769	94	L8
Esters						
**49**	2-Butenoic acid, 3-methyl-, pentyl ester	56922-72-6	C_10_H_18_O_2_	1168	84	L15
**50**	Butyric acid, 1-propylpentyl ester	20286-46-8	C_12_H_24_O_2_	1317	85	L13
**51**	Cyclobutanecarboxylic acid, 2-methylpropanyl ester	87661-19-6	C_9_H_16_O_2_	1141	82	L15
**52**	Cyclobutanecarboxylic acid, 2-pentyl ester	925444-74-2	C_10_H_18_O_2_	1141	84	L3
**53**	Dibutyl phthalate	84-74-2	C_16_H_22_O_4_	2037	83	L3
**54**	1,2-Ethanediol, dipropanoate	123-80-8	C_8_H_14_O_4_	1151	85	L9
**55**	Formic acid, heptyl ester	112-23-2	C_8_H_16_O_2_	1081	89	L4
**56**	(Z)-9-Hexadecen-1-ol acetate	34010-20-3	C_18_H_34_O_2_	1822	81	L4
**57**	Octanoic acid, ethyl ester	106-32-1	C_10_H_20_O_2_	1183	89	L17
**58**	Octanoic acid, pentyl ester	638-25-5	C_13_H_26_O_2_	1481	88	L4
**59**	Oxalic acid, butyl propyl ester	26404-30-8	C_9_H_16_O_4_	1250	87	L9
**60**	2-Phenylacetic acid,2-ethylhexyl ester	5421-30-7	C_16_H_24_O_2_	1758	88	L4
**61**	2-Propenoic acid, tridecyl ester	2495-25-2	C_17_H_32_O_2_	1814	90	L14
Alkenes						
**62**	trans-α-Bergamotene	13474-59-4	C_15_H_24_	1430	81	L11
**63**	3,7-Decadiene	72015-36-2	C_10_H_18_	1032	90	L4
**64**	Decahydro-1,1,4,7-tetramethyl-1H-cycloprop[e]azulene	6790-78-9	C_15_H_26_	1380	83	L8
**65**	3,4-Dimethylpent-1-ene	7385-78-6	C_7_H_14_	1030	84	L4
**66**	(E)-7,11-Dimethyl-3-methylene-1,6,10-dodecatriene	18794-84-8	C_15_H_24_	1440	82	L11
**67**	3,3-Dimethyl-1-octene	74511-51-6	C_10_H_20_	921	88	L1
**68**	1,5-Dodecadiene	84348-04-9	C_12_H_22_	1212	89	L3
**69**	1-Ethoxy-4,4-dimethyl-2-pentene	55702-60-8	C_9_H_18_O	915	84	L11
**70**	8-Heptadecene	16369-12-3	C_17_H_34_	1719	89	L10
**71**	1-Heptadecyne	26186-00-5	C_17_H_32_	1709	92	L10
**72**	10-Heneicosene	95008-11-0	C_21_H_42_	2117	93	L16
**73**	1,15-Hexadecadiene	21964-51-2	C_16_H_30_	1592	91	L6
**74**	1,2,3,5,6,7,8,8a-Octahydro-1,4-dimethyl-7-(1-methylethenyl)-azulene	489-81-6	C_15_H_24_	1490	81	L11
**75**	7-Oxabicyclo[2.2.1]hept-5-ene-2,3-dicarboxylic anhydride	6118-51-0	C_8_H_6_O_4_	1248	81	L6
**76**	(Z)-5-Tetradecene	41446-62-2	C_14_H_28_	1421	90	L13
**77**	3,7,7-Trimethyl-11-methylenespiro[5.5]undec-2-ene	15401-86-2	C_15_H_24_	1507	83	L11
Alkanes						
**78**	1-Butyl-2-ethylcyclopentane	72993-32-9	C_11_H_22_	999	84	L15
**79**	1-Cyclopropylpentane	2511-91-3	C_8_H_16_	819	82	L13
**80**	1,1-Dimethyl-3-methylidene-2-prop-2-enylidenecyclohexane	99647-15-1	C_12_H_18_	788	83	L2
**81**	3,7-Dimethyl-nonane	17302-32-8	C_11_H_24_	986	94	L7
**82**	1,2-Epoxydodecane	2855-19-8	C_12_H_24_O	1304	91	L4
**83**	5-Ethylundecane	17453-94-0	C_13_H_28_	1249	94	L11
**84**	3-Methyl-5-propylnonane	31081-18-2	C_13_H_28_	1185	92	L16
**85**	(S)-{[4-(Phenylmethoxy)phenoxy]methyl}-oxirane	122797-04-0	C_16_H_16_O_3_	410	88	L17
**86**	n-Nonylcyclohexane	2883-02-5	C_15_H_30_	1576	89	L10
**87**	cis-2-Phenyl-1-(2-methyl-1-propenyl)cyclopropane	89486-56-6	C_13_H_16_	1078	83	L3
**88**	Propyl-cyclopropane	2415-72-7	C_6_H_12_	620	88	L2
**89**	2,2,3,3-Tetramethylhexane	13475-81-5	C_10_H_22_	846	97	L19
**90**	(S)-2-Tridecyloxirane	96938-07-7	C_15_H_30_O	1603	84	L19
Furans						
**91**	Dibenzofuran	132-64-9	C_12_H_8_O	1483	90	L20
**92**	Furan	110-00-9	C_4_H_4_O	553	96	L6
**93**	2-Hexyl-2-methyl-5-(propan-2-ylidene)tetrahydrofuran	124099-79-2	C_14_H_26_O	1147	91	L4
**94**	Octahydro-2,3’-bifuran	73373-15-6	C_8_H_14_O_2_	1079	87	L1
Phenols						
**95**	4-Ethyl-2-methoxy-phenol	2785-89-9	C_9_H_12_O_2_	1303	85	L20
**96**	2-Ethylphenol	90-00-6	C_8_H_10_O	1114	89	L20
**97**	2-Methoxy-4-methyl-phenol	93-51-6	C_8_H_10_O_2_	1203	89	L20
**98**	2,4-bis(1,1-dimethylethyl)-Phenol	96-76-4	C_14_H_22_O	1555	86	L8

**Table 3 molecules-27-00308-t003:** Geographical ecological factors of different sampling sites.

Samples	Collected Location	Cultivars	Latitude	Longitude	Altitude (m)	Annual Average Temperature (°C)	Annual Rainfall (mm)	Annual Sunshine Duration (h)	Climate
L1	Sihui, Zhaoqing, Guangdong	*Camellia semiserrata* Chi	23°35′ N	112°33′ E	≤1000	20–22	1750	1600	Subtropical climate
L2	Lianzhou, Qingyuan, Guangdong	*Camellia meiocarpa* Hu	25°05′ N	112°37′ E	≤1000	19–21	1625	1510	Central Asia monsoon climate
L3	Qingxin, Qingyuan, Guangdong	*Camellia oleifera* Abel	23°44′ N	113°0′ E	≤1000	19–21	1625	1510	Central Asia monsoon climate
L4	Yangchun, Yangjiang, Guangdong	*Camellia oleifera* Abel	22°19′ N	111°51′ E	≤200	21–28	2380	2000	Subtropical rainforest climate
L5	Qujiang, Shaoguan, Guangdong	*Camellia oleifera* Abel	24°42′ N	113°49′ E	≤200	18–26	1700	1660	Subtropical monsoon climate
L6	Guangning, Zhaoqing, Guangdong	*Camellia oleifera* Abel	23°39′ N	112°21′ E	≤300	20–22	1720	1613	Transitional climate between South Asia and central subtropics
L7	Yunan, Yunfu, Guangdong	*Camellia oleifera* Abel	22°56′ N	111°53′ E	≤1000	20–25	1580	1480	Subtropical monsoon climate
L8	Longchuan, Heyuan, Guangdong	*Camellia oleifera* Abel	24°19′ N	115°15′ E	≤500	18–27	1500	1700	Subtropical monsoon climate
L9	Xingning, Meizhou, Guangdong	*Camellia meiocarpa* Hu	24°25′ N	115°37′ E	≤400	19–26	1520	1900	Transitional climate between South Asia and central subtropics
L10	Tianhe, Guangzhou, Guangdong	*Camellia gauchowensis* Change	23°11′ N	113°22′ E	≤100	20–28	2000	1620	Subtropical marine monsoon climate
L11	Xuwen, Zhanjiang, Guangdong	*Camellia gauchowensis* Change	20°19′ N	110°19′ E	≤100	20–25	2000	2100	Tropical monsoon climate
L12	Gaozhou, Maoming, Guangdong	*Camellia gauchowensis* Change	21°42′ N	110°36′ E	≤1600	20–25	1900	1950	Subtropical monsoon climate
L13	You, Zhuzhou, Hunan	*Camellia oleifera* Abel	26°46′ N	113°09′ E	≤1400	16–18	1400	NF	Mid-subtropical humid monsoon climate
L14	Yuanzhou, Yichun, Jiangxi	*Camellia oleifera* Abel	27°33′ N	113°54′ E	≤1800	15–20	1680	1740	Mid-subtropical monsoon climate
L15	Zhanggong, Ganzhou, Jiangxi	*Camellia oleifera* Abel	24°29′ N	113°54′ E	300–500	18–22	1320	NF	Subtropical monsoon climate
L16	Qingshanhu, Nanchang, Jiangxi	*Camellia oleifera* Abel	28°10′ N	115°27′ E	≤1000	17–18	1650	1800	Subtropical monsoon climate
L17	Xixiangtang, Nanning, Guangxi	*Camellia gauchowensis* Change	22°48′ N	108°22′ E	300–600	20–23	1300	NF	Subtropical monsoon climate
L18	Xiuying, Haikou, Hainan	*Camellia vietnamensis* Huang ex Hu	19°31′ N	110°24′ E	≤100	27–29	2040	2160	Tropical monsoon climate
L19	Beireng, Qionghai, Hainan	*Camellia vietnamensis* Huang ex Hu	18°58′ N	110°7′ E	≤100	27–28	2040	2155	Tropical monsoon climate
L20	Panlong, Kunming, Yunnan	*Camellia oleifera* Abel	25°02′ N	102°42′ E	1500–2800	13–18	1035	2200	Subtropical highland monsoon climate

Note: NF, not found. Geographic information parameters were from China statistical yearbook sharing platform, www.yearbookchina.com (last accessed on 27 November 2021).

## Data Availability

Data is contained within the article or supplementary material.

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
