# Peer review of "Characterization of the Volatile Compounds in Camellia oleifera Seed Oil from Different Geographic Origins"

_molecules, 2022, doi:10.3390/molecules27010308_

Round 1

Reviewer 1 Report

Major revision comments

  1. Abstract: the aim of the study should be stated in the abstract.
  2. Materials and methods: It is not clear if the methods for sample preparation (oil extraction), HS-SPME, and GC-MS are “in house” methods or the authors used already published methods. If the methods were taken from already published papers, a detailed citation is needed. If they are “in house” methods, the validation parameters should be given.
  3. Results:

Table 2. There are no data regarding the percentage of similarity for aroma compounds No 20 to No 98. An explanation is needed.

Technical issues:

  1. Legends and corresponding explanations should be added to all figures and tables.

Author Response

Major revision comments

1. Abstract: the aim of the study should be stated in the abstract.

Response 1: Thanks for reviewer’s comments. The second sentence in Abstract is the aim of this study in our manuscript (Lind 13-15).

Line 13-15: The purpose of this investigation was to characterize the virgin Camellia oleifera seeds oil (VCO) samples from different locations in the Southern China in terms of their aroma to show the classification of VCO with respect to geography.

2. Materials and methods: It is not clear if the methods for sample preparation (oil extraction), HS-SPME, and GC-MS are “in house” methods or the authors used already published methods. If the methods were taken from already published papers, a detailed citation is needed. If they are “in house” methods, the validation parameters should be given.

Response 2: In Line 325-333 of manuscript with revision mode, a more detailed description about sample preparation has been modified in 3.2 Oil extraction. The analysis methods of volatile compounds by HS-SPME and GC-MS has been studied in our previous research (Kesen et al. 2014) and published as our paper (Wang et al. 2020). And it has been cited into 3.3.1 HS-SPME (Line 336-337).

Line 326-333: After sun exposure and manual shelling, only no infection and physically damaged C. oleifera seeds were obtained from fruits and put into oven at 75 for hot air drying until constant weight [47,48]. Then, the seeds were crushed and transferred to a BOZY-01G screw press from Hanhuang Electric Appliance Technology Co., Ltd (Zhejiang, China). After the screw-pressing process, the crude oil was centrifuged at 10,000 rpm for 10 min at 4 . Finally, the supernatant oil as virgin C. oleifera seeds oil (VCO) was kept in brown glass bottles in cool place until analysis.

Line 336-337: The volatile compounds analysis of VCO was based on the previous flavor research [18] and further optimized by ourselves [49].

Kesen, S.; Kelebek, H.; Selli, S. Characterization of the key aroma compounds in Turkish olive oils from different geographic origins by application of aroma extract dilution analysis (AEDA). Agr. Food Chem. 2014, 62, 391–401. [https://doi.org/10.1021/jf4045167]

Wang, J.; Tang, X.; Zhang, Y.; Wang, M.; Xu, B.; Zhang, M. Optimization of headspace solid phase micro-extraction of volatile components from Camellia oleifera seeds oil by response surface methodology. Food Res. Dev. 2020, 41, 93-100. [https://doi.org/10.12161/j.issn.1005-6521.2020.13.015]

3. Results: Table 2. There are no data regarding the percentage of similarity for aroma compounds No 20 to No 98. An explanation is needed.

Response 3: Sorry for missing the percentage of similarity for aroma compounds No 20 to No 98. Thanks for reviewer’s reminding. It has been completed all the information in Table 2 (Line 242).

Technical issues:

4. Legends and corresponding explanations should be added to all figures and tables.

Response 4: Thanks for reviewer’s kindly suggestion. All legends and corresponding explanations in each figure and table have been checked through and added, such as Figure 2 in Line 180-184 of manuscript with revision mode.

Line 180-184: Figure 2. The compositions and contents of volatile compounds of VCO from 20 geographic regions. A) The number of ketones, alcohols, alkanes, esters, aldehydes, alkenes, acids, phenols, furans, and benzenes in all Camellia oleifera seeds oils. B) The relative contents of ten chemical categories of aroma in VCO samples from 20 regions. C) The number of various volatile components in VCO samples from 20 regions.

Merry Christmas and Happy New Year!

Reviewer 2 Report

Revision
Characterization of the Volatile Compounds in Camellia oleifera seeds oil from Different
Geographic Origins
The topic is interesting and not explored in scientific journals, thus may gather many readers and brings substantial amount of citations. However in some parts should be clarified, supplemented or revised.
Introduction:
1. In Introduction Authors not explained (not justified) why the Camellia oleifera seeds oil deserves attention to be the subject of this study (with regard fatty acid composition, phenolic compounds etc... or/and properties). Please supplement.
2. In lines 63-70 should be included more information and citation about using profile of volatile compounds for discrimination of geographical origin (site, location) oils, fruits ... origin. Please insert the following citations
Mozhgan Zarifikhosroshahi, Zehra Tugba Murathan, Ebru Kafkasa, Volkan Okatan. Variation in volatile and fatty acid contents among Viburnum opulus L. fruits growing different location, Scientia Horticulturae 264 (2020). https://doi.org/10.1016/j.scienta.2019.109160
Tabaszewska M., Antoniewska A., Rutkowska J., Skoczylas Ł., Słupski J., Skoczeń-Słupska R. Bioactive components, volatile profile and in vitro antioxidative properties of Taxus baccata L. red arils. Molecules 2021, 26(15), 4474; https://doi.org/10.3390/molecules26154474
The part in lines 63-70 should be extended.
3. Lines 58-62: I suggest to delete redundant sentences (it si commonly known).
4. I suggest to reformulate the aim. For example it should explained, what was the reason to study volatile compounds in samples of Camellia oleifera oil derived from twenty locations/sites ?
Results and Discussion:
1. I recommend to supplement discussion of obtained results in the part 2.2. There were no sufficient comparison of volatile profile of studied oil with other oils and other vegetable sources with regard geographical origin.
2. In table 1 I am not sure that all columns are necessary. Please consider that ?
General comment:
The authors conducted advanced statistical analysis however, the implications of this have not been explained. For example the sentence in abstract in Lines 25-30 is not informative. How geographical and climate conditions in the studied regions influenced volatile profile of oil sample (please insert
some information in abstract and conclusions).

Author Response

Revision

Characterization of the Volatile Compounds in Camellia oleifera seeds oil from Different Geographic Origins

The topic is interesting and not explored in scientific journals, thus may gather many readers and brings substantial amount of citations. However, in some parts should be clarified, supplemented or revised.

Introduction:

1. In Introduction Authors not explained (not justified) why the Camellia oleifera seeds oil deserves attention to be the subject of this study (with regard fatty acid composition, phenolic compounds etc... or/and properties). Please supplement. (Point 1)

Response 1: Thanks very much for reviewer’s high affirmation and professional comments for this manuscript. As the reviewer’s suggestion, because of its fatty acid composition, polyphenols, squalene, sterol and vitamin E, Camellia oleifera seeds oil is a woody vegetable oil with great research value. It has been added into 1. Introduction (Line 43-47 in manuscript with revision mode).

Line 43-47: C. oleifera seeds oil contains squalene [3], sterol [3-5], tocopherol [5], polyphenols [6], and a high content (90%) of unsaturated fatty acids (mainly oleic acids and linoleic acid) [7-8], and plays important roles in antioxidant [9,10], anti-inflammatory [11], hepatoprotective and gastroprotective functions [12].

2. In lines 63-70 should be included more information and citation about using profile of volatile compounds for discrimination of geographical origin (site, location) oils, fruits ... origin. Please insert the following citations

  • Mozhgan Zarifikhosroshahi, Zehra Tugba Murathan, Ebru Kafkasa, Volkan Okatan. Variation in volatile and fatty acid contents among Viburnum opulus L. fruits growing different location, Scientia Horticulturae 264 (2020). https://doi.org/10.1016/j.scienta.2019.109160
  • Tabaszewska M., Antoniewska A., Rutkowska J., Skoczylas Ł., Słupski J., Skoczeń-Słupska R. Bioactive components, volatile profile and in vitro antioxidative properties of Taxus baccata L. red arils. Molecules 2021, 26(15), 4474; https://doi.org/10.3390/molecules26154474

The part in lines 63-70 should be extended. (Point 2)

Response 2: According to the reviewer’s suggestion, a further introduction of the profile of volatile compounds for discrimination of geographical origin with other oil or fruit has been added into the 1. Introduction. Two of references above also have been cited in corresponding position. The part in Line 63-70 of original manuscript has been modified for extending introduction (Line 72-88 in manuscript with revision mode).

Line 72-88: The flavor composition and nutritional evaluation of edible plant in different growing locations has been an important research field in the past few years, such as focusing on volatile compounds, fatty acids, amino acids, polyphenols, and antioxidant activities in Capsicum annuum [23], Viburnum opulus L. [24], olive [18, 25-27], Taxus Baccata L. [28], Paeonia ostii [29], and Camellia sinensis [30]. It has long been known that the aroma profiles of edible oil are related to genetic (cultivars) [31-35], environmental (geography [18,25,27,31], climatic conditions [27,31] and storage conditions [31]), cultivating (agronomic techniques [36] and the degree of fruit ripening [31,36]), and processing (harvesting methods [31] and processing technology [37]) factors. Therefore, geographic origin of C. oleifera is greatly responsible for the sensorial characteristics of VCO. Moreover, the volatile compounds of oil obtained from different production area, under identical growth conditions, harvested at roughly equal ripeness degree, and processed in the same manner, can be characterized by different compositions and their respective concentrations. There has been increasing interest in the geographical identification of virgin plant oil, as a reliable criterion for its authentication and quality [25].

3. Lines 58-62: I suggest to delete redundant sentences (it is commonly known). (Point 3)

Response 3: The introduction about multivariate statistical methods has been deleted as recommended (Line 66-70 in manuscript with revision mode).

4. I suggest to reformulate the aim. For example it should explained, what was the reason to study volatile compounds in samples of Camellia oleifera oil derived from twenty locations/sites ? (Point 4)

Response 4: The aim of this study described in last paragraph of 1. Introduction has been reformulated (Line 93-97 in manuscript with revision mode).

Line 93-97: Therefore, the aim of this study was to investigate the characterization of the volatile compounds extracted from VCO produced in different geographical locations of Southern China by GC-MS with multivariate statistical methods, and to find out the specific volatile substances or their categories that probably affect the VCO flavor from different geographical regions.

Results and Discussion:

5. I recommend to supplement discussion of obtained results in the part 2.2. There were no sufficient comparison of volatile profile of studied oil with other oils and other vegetable sources with regard geographical origin. (Point 5)

Response 5: More discussion about the comparison of volatile profile of studied oil with other oils and other vegetable sources with regard geographical origins has been added into part 2.2 such as in Line 165-170 of manuscript with revision mode.

Line 165-170: In the study of other foods flavor with regard geographical origins, aldehydes and alcohols are considered the largest volatile profiles in olive oil with the 2-hexenal and 1-hexanol more than 50% [18,27] and 12% [25], respectively. Alkanes are main volatile compounds in bell pepper spices [23], and acids and ketones are main volatile compounds in Viburnum opulus L. fruits [24]. Compared to other oils or plants, VCO presents its own special flavor.

6. In table 1 I am not sure that all columns are necessary. Please consider that ? (Point 6)

Response 6: Thanks for reviewer’s comments. The threshold value in Table 1 has been deleted. Moreover, according to other reviewers’ suggestion, a column of each aroma compound CAS has been added into Table 1 and 2.

General comment:

The authors conducted advanced statistical analysis however, the implications of this have not been explained. For example the sentence in abstract in Lines 25-30 is not informative. How geographical and climate conditions in the studied regions influenced volatile profile of oil sample (please insert some information in abstract and conclusions). (Point 7)

Response 7: Thanks again for reviewer’s professional comments. The sentence in Line 25-30 of original manuscript has been modified. A further discussion about the geographic characterization of the volatile compounds in different VCO by multivariate statistical methods has been added in abstract and conclusion (Line 25-33, Line 393-405).

Line 25-33: After PCA, heatmap and PLS-DA analysis, Longchuan of Guangdong (L8), Qingshanhu of Jiangxi (L16), and Panlong of Yunnan (L20) were in one group where the annual average temperatures are relatively low and the annual rainfalls are also low. Guangning of Guangdong (L6), Yunan of Guangdong (L7), Xingning of Guangdong (L9), Tianhe of Guangdong (L10), Xuwen of Guangdong (L11), and Xiuying of Hainan (L18) were in another group where the annual average temperatures are relatively high and the altitudes are low.

Line 393-405: The regions of Longchuan in Guangdong (L8), Qingshanhu in Jiangxi (L16), Panlong in Yunnan (L20) where plant same cultivars of C. oleifera Abel. belonged to the same category, scatter plots of which was in the lower part in partial least squares-discrimination analysis compared to other groups. From the perspective of growing environments, the annual average temperatures in these 3 locations are relatively low (lowest at 13 ) and the annual rainfalls are also low (1035-1650 mm). Moreover, another group contained 6 planting areas of Guangning in Guangdong (L6), Yunan in Guangdong (L7), Xingning in Guangdong (L9), Tianhe in Guangdong (L10), Xuwen in Guangdong (L11), and Xiuying in Hainan (L18) where the annual average temperatures are relatively high (highest at 29 ) but the altitudes are low (1000 m). The influence of geography and climate on VCO flavor is highly complicated. The geographical characterization of the other two groups were not found in our study, probably there are some unknown factors affecting their classification.

Merry Christmas and Happy New Year!

Reviewer 3 Report

My comments in the attached PDF file. 

Author Response

Point 1: Is Figure 1 TIC profile of each sample. Volatile intensities and varieties of each sample may need to consider the sample amount used. (Line 85 in original manuscript)

Response 1: Thanks for reviewer’s comments. Figure 1 showed the total ion chromatograms of volatile compounds of L1 to L20 samples, and L4 sample showed the largest number of peaks from TIC profile. All the samples were made by screw-pressing and belonged to virgin oil (3.2 Oil extraction in 3. Material and Methods). When it was tested, 20 ml for each sample was put into 40 ml headspace vials for extraction (3.3.1 HS-SPME in 3. Material and Methods). Therefore, each sample amount was considered same for Figure 1 and whole manuscript analysis.

Line 330-333 in manuscript with revision mode: After the screw-pressing process, the crude oil was centrifuged at 10,000 rpm for 10 min at 4 . Finally, the supernatant oil as virgin C. oleifera seeds oil (VCO) was kept in brown glass bottles in cool place until analysis.

Line 337-341 in manuscript with revision mode: The 20 ml samples of VCO were put into 40 ml headspace vials, which were hermetically sealed with silicone pad. The headspace vials with oil were allowed to equilibrate for 10 min at room temperature. The target aroma substances of the samples were extracted for 33 min at 40 using a headspace solid phase microextraction manual sampler injection handle.

Point 2: L6 or L16? (Line 87 in original manuscript)

Response 2: Thanks very much for your reminding. Yes, it should be L16 and it has been modified (Line 108-109 in manuscript with revision mode).

Line 108-109: Moreover, L16 displayed longest retention time span and it had an obvious abundance at 47.50 min that was hexatriacontane by system software analysis.

Point 3: Figure 1: resolution not very good. base line of some samples not cleared (i.e. L18- L20) (Line 89 in original manuscript). Figure quality is not good (Line 94 in original manuscript).

Response 3: The quality of Figure 1 has been improved, especially for the resolution and base lines. Wish it is better now.

Point 4: Are these descriptors only from the reference? Any further confirmation tests for supporting of the flavors for the VCO samples? (L160-161 in original manuscript)

Response 4: In the study of food flavor, specific volatile substances have specific threshold and also represent specific flavor. No further test proof is required. In our study, the flavor description of specific volatile compounds is referenced from a published book (Burdock, 2016). In order to better understand, this sentence has been modified and relevant reference has been marked at the corresponding location (Line 190-195 in manuscript with revision mode).

Line 190-195: Among them, decanal with sweety flavor, 2,4-decadienal with deep-fried flavor, (E)-2-decenal with fatty flavor, and 2-undecenal with fresh aldehyde flavor [41], were found in all of VCO samples and their retention indexes were approximate (1204-1311), suggesting that these flavors are highly common in VCO.

Burdock, G.A. Fenaroli's handbook of flavor ingredients, sixth edition. 2016.

Point 5: A column for CAS should be better for compound ID (Table 1) (Line 182). You may need a more column for CAS number of each compound (Table 2) (Line 207 in original manuscript).

Response 5: Each aroma compound CAS has been added into Table 1 and 2. Moreover, according to other reviewers’ suggestion, the threshold value in Table 1 has been deleted.

Point 6: did you use all chemicals discovered in VCO or use the sum of your compound groups for these statistics. I think you should clarify here. (Line 209 in original manuscript)

Response 6: Thanks for reviewer’s professional comments. Ten compound groups of volatile components for ketones, alcohols, alkanes, esters, aldehydes, alkenes, acids, phenols, furans, and benzenes were used for PCA in our study, which has been modified and clarified in 2.3.1. Principal component analysis.

Line 247-250: From the score plots of the principal component analysis (PCA) for ketones, alcohols, alkanes, esters, aldehydes, alkenes, acids, phenols, furans, and benzenes as shown in Figure 3A, all samples were scattered in four parts.

Point 7: You did not clarify what is the key chemicals to make these varieties differed from others. (Line 216-217 in original manuscript)

Response 7: PCA is a kind of analysis method to show the differences and relationships among samples statistically [Wu et al. 2021], which cannot explain the specific substances leading to the difference. Therefore, this sentence (Line 252-253 in manuscript with revision mode) is followed by a small conclusion or reason for this difference (Line 253-255).

Thanks very much for reviewer’s this question. Actually, this is also the aim to analysis Table 1 and 2 in this manuscript, which specific chemicals cause the flavor difference of each sample (Line 208-209, Line 214-215, and Line 229-237).

Line 252-253: The samples that have more specific volatile substances such as L4, L8, and L20 were farther away from the PCA center point.

Line 253-255: Therefore, it could be inferred that the 23 common (Table 1) and 98 unique (Table 2) volatile compounds determine the particularity of VCO flavor in various planting regions.

Line 208-209: It is noteworthy that L8 sample had not common acids that probably implies the specific of L8.

Line 214-215: Maltol (3-hydroxy-2methyl-4-pyrone), which is usually found in roasted cocoa powder as the volatile flavor compound of caramel [28], was also not recognized in L8 sample.

Line 229-237: L4 had the largest number of unique volatile compounds, indicating that its characteristic flavor was probably different from other samples. The unique volatile compounds of L4 were mostly concentrated in acids and esters that shows the specific aroma of flowers, iris, and fruit (octanoic acid pentyl ester). Besides, the characteristic flavor of L8 sample probably is weak floral, woody, and slight scented with laurel oil (dodecanoic acid). The characteristic aroma of L20 sample might be herbs (4-ethyl-2-methoxy-phenol), spicy (2-methoxy-4-methyl-phenol), and phenol odor (2-ethyl-phenol). Meanwhile, no specific compounds were detected in L12 and L18, demonstrating that the flavor of these two samples may be more popular and have few characteristics.

Wu, J.; Fan, X.; Huang, X.; Li, G.; Guan, J.; Tang, X.; Qiu, M.; Yang, S.; Lu, S. Effect of different drying treatments on the quality of camellia oleifera seed oil. S. Afr. J. Chem. Eng. 2021, 35, 8–13. [https://doi.org/10.1016/j.sajce.2020.10.003]

Point 8: Table 3? unique volatile compounds? something wrong here. Do you mean Table 2?  But only 98 compounds listed in Table 2. (Line 218 in original manuscript)

Response 8: It should be Table 2 and 98 compounds, which has been modified.

Line 253-255: Therefore, it could be inferred that the 23 common (Table 1) and 98 unique (Table 2) volatile compounds determine the particularity of VCO flavor in various planting regions.

Point 9: combined or collected (Line 284 in original manuscript)

Response 9: “gathered” has been revised to “collected” in Line 319 of manuscript with revision mode.

Point 10: maturity stage does influence in fruit flavour. This is very cleard in the fruit such as apple kiwifruit. Do you have a standard giving your sample "almost the same maturity stage"? such as colour, size, firmness, soluble solid, brix..... (Line 285 in original manuscript)

Response 10: Every year, after Frost’s Descent that is a kind of the 24 traditional Chinese solar terms, it is commonly considered that Camellia oleifera is mature. After that day, Camellia oleifera fruits would be widely harvested all over the country, and it usually lasts about one month. Because of the wide collection regions of Camellia oleifera seeds in this study (Table 3), it is hard to fix in a few days to pick and collect all the samples. However, from the dehiscent degree of fruit and brown of seed, it is certain that they are all in the same mature stage. Our appearance of fruit and seed was the same with Zhu et al. (2019) report (Figure 1 in their paper).

Zhu, G.; Liu, H.; Xie Y.; Liao Q.; Lin Y.; Liu Y.; Liu Y.; Xiao H.; Gao Z.; Hu S. Postharvest processing and storage methods for Camellia oleifera seeds. Food Rev. Int. 2019, 1–21. [https://doi.org/10.1080/87559129.2019.1649688]

Merry Christmas and Happy New Year!

Round 2

Reviewer 2 Report

I am happy with the revision and answers given, of particular addressed the necessary explanation of differences in volatile profile of Camellia oleifera seed oil regarding geographical origin. Good luck in your future work.

Author Response

I am happy with the revision and answers given, of particular addressed the necessary explanation of differences in volatile profile of Camellia oleifera seed oil regarding geographical origin. Good luck in your future work.

Response:

Thanks very much for reviewer’s affirmation and blessing of our work.

It is hoped that the reviewer would agree to publish this manuscript on Molecules.

Again, Happy New Year and Best Regards!